# Investigating the Role of Heparanase in Breast Cancer Development Utilising the MMTV-PyMT Murine Model of Mammary Carcinoma

**DOI:** 10.3390/cancers15113062

**Published:** 2023-06-05

**Authors:** Krishnath M. Jayatilleke, Hendrika M. Duivenvoorden, Gemma F. Ryan, Belinda S. Parker, Mark D. Hulett

**Affiliations:** 1Department of Biochemistry and Chemistry, La Trobe Institute for Molecular Science, La Trobe University, Melbourne, VIC 3086, Australia; krish.jayatilleke@precisionformedicine.com (K.M.J.); hendrika.duivenvoorden@monash.edu (H.M.D.); gemma.ryan@latrobe.edu.au (G.F.R.); belinda.parker@petermac.org (B.S.P.); 2School of Biological Sciences, Monash University, Clayton, VIC 3168, Australia; 3Sir Peter MacCallum Department of Oncology, The University of Melbourne, Parkville, VIC 3052, Australia

**Keywords:** heparanase, breast cancer, MMTV-PyMT

## Abstract

**Simple Summary:**

Heparanase (HPSE) has been demonstrated to enhance the progression and metastasis of solid tumours, leading to a poor clinical prognosis. However, robust genetic ablation animal models to investigate the role of HPSE in breast cancer have not been described. The aim of this study was to utilise an HPSE-deficient strain of the well-established MMTV-PyMT murine mammary carcinoma model (MMTV-PyMTxHPSE^−/−^ mice) to investigate the role of HPSE in early establishment, progression, and metastasis of mammary tumours. Contrary to our current understanding, we observed that even though HPSE regulated tumour angiogenesis, the establishment, progression, and metastasis of mammary tumours in MMTV-PyMT animals were HPSE-independent. We further observed no compensation by matrix metalloproteinases in response to the lack of HPSE in MMTV-PyMTxHPSE^−/−^ animals. These findings may have significant implications in the development and clinical utility of HPSE inhibitors.

**Abstract:**

Breast cancer is the second most common human malignancy and is a major global health burden. Heparanase (HPSE) has been widely implicated in enhancing the development and progression of solid tumours, including breast cancer. In this study, the well-established spontaneous mammary tumour-developing MMTV-PyMT murine model was utilised to examine the role of HPSE in breast cancer establishment, progression, and metastasis. The use of HPSE-deficient MMTV-PyMT (MMTV-PyMTxHPSE^−/−^) mice addressed the lack of genetic ablation models to investigate the role of HPSE in mammary tumours. It was demonstrated that even though HPSE regulated mammary tumour angiogenesis, mammary tumour progression and metastasis were HPSE-independent. Furthermore, there was no evidence of compensatory action by matrix metalloproteinases (MMPs) in response to the lack of HPSE expression in the mammary tumours. These findings suggest that HPSE may not play a significant role in the mammary tumour development of MMTV-PyMT animals. Collectively, these observations may have implications in the clinical setting of breast cancer and therapy using HPSE inhibitors.

## 1. Introduction

Heparanase (HPSE) is a β-D-endoglucuronidase and is the only mammalian enzyme capable of cleaving heparan sulphate (HS), a key extracellular matrix (ECM) and basement membrane (BM) component, leading to ECM and BM remodelling [1,2]. The enzymatic activity of HPSE alters the structural integrity of the ECM and BM, thereby promoting cell invasion and enhancing the bioavailability of HS-bound growth-promoting and signalling molecules, termed HS-binding proteins (HSBPs) [3,4,5]. This drives both physiological and pathological processes such as angiogenesis, cell migration, inflammation, and metastasis [6,7,8,9,10,11,12,13,14]. HPSE is naturally expressed only in limited cell and tissue types such as immune cells, the placenta, and platelets [1,6,9,15,16]. However, dysregulated gene expression in malignant cells leads to overexpression of HPSE in all cancers [17]. By virtue of its HSBP-modulating activity, the expression of HPSE within the tumour microenvironment (TME) leads to enhanced cancer growth and metastasis and essentially upregulates all hallmark features of cancer [18]. Over four decades of research have cemented the key role of HPSE in malignant disease progression and it has emerged as an attractive but challenging anti-cancer therapeutic target [19,20].

Breast cancer was the most commonly diagnosed cancer in 2020 and led to 6.9% of all cancer-related deaths [21]. An early understanding of the role of HPSE in breast cancer arose from demonstrating that highly metastatic MDA-MB-435 human breast cancer cells expressed a high level of HPSE in contrast to the moderately and weakly metastatic MDA-MB-231 and MCF-7 breast cancer cells, respectively [2]. Early in vivo studies demonstrated a correlation between HPSE expression and resultant tumour growth and angiogenesis [22]. A methylation-dependent regulation of HPSE expression which correlates directly with tumour aggressiveness was demonstrated with the use of breast cancer patient samples [23]. HPSE expression was shown to be regulated by estrogen in both in vitro and in vivo settings, suggesting its hormonal-driven regulation in breast cancer [24]. Recently, HPSE was shown to enable circulating tumour cells clusters by regulating FAK-1 and ICAM-1-mediated cell adhesion, enhancing breast cancer metastasis [25]. Murine breast cancer was significantly enhanced with the expression of HPSE and was further promoted with the co-expression of mutant *H-Ras* [26]. The mouse mammary tumour virus (MMTV) regulatory element-driven expression of the C-terminal domain of HPSE also induced mammary tumour growth [27]. Resistance to lapatinib in the human brain-colonising MDA-MB-231BR breast cancer cell line was attenuated through the inhibition of HPSE by SST0001 (Roneparstat) [28]. Overexpression of extracellular superoxide dismutase was shown to inhibit HPSE expression, which in turn reduced in vitro breast cancer cell growth and invasion [29]. The mode of action of the plant extract elemene, a traditional Chinese anti-cancer remedy, had remained elusive for decades [30]. Recently, by using the 4T1 murine breast cancer cells, it was demonstrated that elemene was capable of downregulating HPSE, thus imparting anti-proliferative and anti-metastatic effects. Proliferation, primary tumour growth, angiogenesis, and metastasis of the highly aggressive MDA-MB-435 human breast cancer cell-derived tumours was attenuated by the use of siH1324, an siRNA acting as an *HPSE* gene-specific inhibitor [31]. PG545, an HS-mimetic and HPSE inhibitor, significantly reduced 4T1 mammary tumour growth and metastasis in mice [32].

Numerous patient-derived data have explored the relationship between HPSE and breast cancer. Recently, a meta-analysis demonstrated that HPSE expression was upregulated in most human breast cancer specimens and was associated with larger tumours, metastasis, histological tumour grade, and poor survival [33]. HPSE expression was associated with reduced HS deposition and increased metastatic potential of breast cancer in patient samples [34]. The invasive capacity of ductal carcinoma in situ (DCIS) lesions directly correlated with their expression of HPSE [35]. HPSE and COX-2 were shown to be predictive markers of lymph node metastasis in high-grade breast tumours [36]. The micro RNA species miR-1258 was identified to directly target HPSE and inhibit brain-metastatic breast cancer [37]. In support of HPSE-mediated breast cancer metastasis, miR-1258 levels were shown to inversely correlate with HPSE expression and activity and the overall metastatic potential of breast cancer cells. HPSE was shown to play a role in therapy resistance in breast cancer when tamoxifen was suggested to induce HPSE expression in estrogen receptor (ER)-positive breast cancer cells, thus imparting a growth advantage [38]. This was shown to occur through the recruitment of AIB1, an ER co-activator, to the *HPSE* promoter, resulting in a tamoxifen-mediated agonistic effect on HPSE expression. This finding may explain the failure of tamoxifen therapy in certain breast cancer patients. Recent clinical trial data have strongly implicated HPSE leading to poor clinical prognosis in ER-positive breast cancer patients [39]. Triple-negative breast cancer in metastasis was demonstrably enhanced by HPSE activity [40].

Despite decades of research, much remains unknown regarding the precise role of HPSE in the establishment, early progression, and metastasis of breast cancer, largely due to the lack of robust in vivo models. To date, in vivo models for studying the role of HPSE in cancer have focused on using transgenic HPSE-overexpressing mice and immunodeficient mice. The use of HPSE-overexpressing mice often is not a true representation of the cancer setting, and it is preferable to use an HPSE-deficient model in comparison with wild type animals for such studies. In addition, the immune system has been demonstrated to play a key role in cancer development and HPSE is now known to regulate key components of the immune system [6,7,12,13]. Therefore, the use of immunodeficient mice to study the effects of HPSE in cancer also has limitations. Furthermore, there are no reported studies utilising a mouse model of spontaneous mammary tumour growth. To address these gaps in the literature, spontaneous mammary tumour-developing MMTV-PyMT mice devoid of HPSE expression (MMTV-PyMTxHPSE^−/−^) were used in the study of breast cancer. Here, we investigate the development of mammary tumours in both MMTV-PyMT and MMTV-PyMTxHPSE^−/−^ mice to determine the long-term effects of HPSE in breast cancer growth in a spontaneous mammary tumour model.

## 2. Materials and Methods

### 2.1. Generation of MMTV-PyMTxHPSE^−/−^ Mice

All in vivo animal procedures were performed in compliance with the Australian National Health and Medical Research regulations and were reviewed and approved by the La Trobe University Animal Ethics Committee. C57Bl/6xHPSE^−/−^ mice were generated as previously described [7]. To generate MMTV-PyMTxHPSE^−/−^ mice, male MMTV-PyMT mice (on a C57Bl/6 genetic background) were crossed with female C57Bl/6xHPSE^−/−^ mice (Appendix A). Male MMTV-PyMTxHPSE^+/−^ mice were then crossed with female C57Bl/6xHPSE^+/−^ mice. This gave rise to progeny of six distinct genetic backgrounds: C57Bl/6, C57Bl/6xHPSE^−/−^, C57Bl/6xHPSE^+/−^, MMTV-PyMT, MMTV-PyMTxHPSE^−/−^, and MMTV-PyMTxHPSE^+/−^. Of these, C57Bl/6, C57Bl/6xHPSE^−/−^, MMTV-PyMT, and MMTV-PyMTxHPSE^−/−^ animals were selected for further breeding. Male MMTV-PyMT or MMTV-PyMTxHPSE^−/−^ mice were crossed with female C57Bl/6 or C57Bl/6xHPSE^−/−^ mice, respectively, as required. For ethical reasons, female MMTV-PyMT and MMTV-PyMTxHPSE^−/−^ mice were not used in mating as these animals would develop spontaneous mammary tumours, hindering their nursing capability.

### 2.2. Genotyping Strategy

Mouse ear clippings at 21 days of age were digested at 95 °C in 50 mM NaOH with frequent mixing for 12–15 min and promptly neutralised with 1/6 volumes of Tris-HCl (1.0 M, pH 8.0). PCR was performed using GoTaq^®^ Green master mix (M7123, Promega, Madison, WI, USA) as per manufacturer’s instructions. Genotyping was carried out using four primer pairs, as in Table 1, with one cycle of 95 °C for 5 min, 35 cycles of 95 °C for 30 s, 60 °C for 30 s, 72 °C for 25 s, and a final extension cycle of 72 °C for 10 min.

### 2.3. HPSE Enzymatic Activity Assay

A time-resolved fluorescence energy transfer (TR-FRET)-based assay was employed. Samples were diluted at a 1:1 ratio in buffer (20 mM Tris-HCl, 0.15 M NaCl, 0.1% CHAPS, pH 5.5) followed by the addition of Biotin-HS-Eu(K) (0.7 μg/mL Biotin-HS-Eu(K), 0.2 M NaCH_3_CO_2_, pH 5.5) with the reaction incubated at 37 °C for 2 h. Streptavidin-conjugated XL665 (1 μg/mL Strepavidin-XL665, 0.1 M NaPO_4_, pH 7.5, 1.2 M KF, 0.1% (*w*/*v*) BSA, 2.0 mg/mL heparin) was then added, followed by incubation in the dark for 16 h at room temperature. Using a spectrophotometer, excitation at 315 nm and emission at both 620 and 668 nm were measured. The percentage of HS degradation was calculated relative to FRET-negative or -positive samples (absence or presence of XL665-conjugated Streptavidin in the absence of purified HPSE).

### 2.4. Measurement of HPSE Activity of Mouse Splenic Lysate

Animals at 10–12 weeks of age were euthanised with the spleen rapidly harvested and stored in ice-cold RPMI medium (11875093, ThermoFisher Scientific, Waltham, MA, USA) supplemented with 10% (*v*/*v*) foetal calf serum (Interpath, Somerton, VIC, Australia). The spleen was punctured at either end with a 23 G needle, and the splenic cellular content was flushed out with phosphate-buffered saline (PBS). The cells were passed through a 70 μm strainer and collected at 400× *g* for 4 min at 4 °C. The cell pellet was resuspended in red blood cell (RBC) lysis buffer (155 mM NH_4_Cl, 12 mM NaHCO_3_, 0.1 mM EDTA) and incubated for 5 min at room temperature. Lysed RBCs were removed at 400× *g* for 4 min at 4 °C. The resulting cell pellet was lysed by resuspending in Cytobuster^TM^ protein extraction reagent (Merck, Rahway, NJ, USA) at 4 °C as per manufacturer’s instructions. Cellular debris was then separated at 16,000× *g* for 10 min at 4 °C. The supernatant was collected and subjected to protein quantification as per manufacturer’s instructions (Pierce^TM^ BCA protein assay kit, ThermoFisher scientific). Equal amounts of total cellular protein were used in the HPSE enzymatic activity assay.

### 2.5. Purification of Human HPSE

HPSE was purified from human platelets as described previously [41].

### 2.6. Validation of Purified HPSE by Western Blot

Purified HPSE was loaded onto a NuPAGE^TM^ 4–12% Bis-Tris pre-cast protein gel (ThermoFisher Scientific), resolved, and transferred onto a Nitrocellulose membrane (GE Healthcare, Chicago, IL, USA) using an XCell Sure Lock^TM^ electrophoresis system (ThermoFisher Scientific) in NuPAGE transfer buffer (ThermoFisher Scientific). The membrane was then blocked in 5% (*w*/*v*) skim milk in PBS for 1 h at room temperature. Incubation with the primary anti-HPSE antibody (2 μg/mL, rabbit polyclonal, AB85543, Abcam, Cambridge, UK) was performed at 4 °C for 16 h in 5% (*w*/*v*) skim milk in 0.1% (*v*/*v*) Tween-20/PBS. The membrane was then washed in 0.1% (*v*/*v*) Tween-20/PBS, three times. The bound primary antibody was detected by using a secondary donkey anti-rabbit IgG (0.5 μg/mL, NA934, GE Healthcare) in 5% (*w*/*v*) skim milk in 0.1% (*v*/*v*) Tween-20/PBS for 1 h at room temperature. Chemiluminescence detection was then performed using the Pierce ECL Western blotting substrate (ThermoFisher scientific).

### 2.7. Mammary Tumour Measurements

Mouse mammary tumours were measured 2–3 times weekly with the use of electronic callipers and employing the formula (length × width^2^)/2 mm^3^. A total cumulative tumour volume of ≥1500 mm^3^ was considered the ethical end point.

### 2.8. Dissection of Mammary Glands and Mammary Tumours

Female MMTV-PyMT and MMTV-PyMTxHPSE^−/−^ mice were humanely euthanised with a manual cervical dislocation. Animals were pinned in a supine position followed by performing a vertical incision from the base of the tail to the base of the neck. The skin was separated from its underlying connective tissue using a blunt dissection technique and was peeled back and away from the peritoneal cavity. The mammary glands are located on the inner surface of the skin: the cervical (1st), thoracic (2nd and 3rd), and inguinal (4th and 5th), totalling ten mammary glands per mouse (Appendix A). The 2nd and 3rd thoracic mammary glands are indistinct from one another and were therefore excised together as one for analysis.

### 2.9. Whole Mounting of Mouse Mammary Glands

Excised mouse mammary glands were laid outstretched and flattened on a glass slide (SuperFrost^TM^, Thermo Scientific). The slide with the mammary gland was then incubated in Canoy’s fixative (a solution of ethanol/chloroform (Merck)/acetic acid (Merck) in a 6:3:1 ratio, respectively) for 16 h at room temperature. The slide was then rehydrated by incubating in 70% ethanol, 15 min; 50% ethanol, 15 min; 20% ethanol, 15 min; and distilled water, 5 min. The tissue was then stained in Carmine red (a solution of 0.2% (*w*/*v*) Carmine red (Sigma-Aldrich, St. Louis, MO, USA) and 0.5% (*w*/*v*) aluminium potassium sulphate (Sigma-Aldrich), boiled for 20 min, filtered, with a crystal of thymol (Sigma-Aldrich) added as a preservative, then stored at 4 °C) for 16 h at room temperature. The tissue was then dehydrated by incubating in 70% ethanol, 15 min; 90% ethanol, 15 min; and 100% ethanol, 15 min and stored in histolene (Grale, Ringwood, VIC, Australia) for photography under a light microscope (Appendix A).

### 2.10. H&E Staining

The staining protocol was as follows: histolene (Grale) 3 min (three times); 100% ethanol, 1 min (three times); 70% ethanol, 30 s; distilled water, 1 min; haematoxylin, 4 min; distilled water, 1 min (twice); Scotts tap water substitute (Amber Scientific, Midvale, WA, Australia), 45 s; distilled water, 2 min; eosin (Amber Scientific), 4 min; distilled water, 15 s; 100% ethanol, 45 s (three times); and histolene, 3 min (three times). Mounting medium (Entellan, Merck) was added on to the section prior to placing cover slips (Menzel Gläser, Braunschweig, Germany). The slides were then dried for 16 h at room temperature.

### 2.11. Immunohistochemistry (IHC)

Paraffin section slides were first de-waxed as follows: histolene (Grale), 4 min (three times); 100% ethanol, 1 min (three times); and 70% ethanol, 1 min, and then re-hydrated in distilled water. As required, antigen retrieval was performed in a de-cloaking chamber (Biocare Medical, Pacheco, CA, USA) for 5 min at 5 psi and at 110 °C in sodium citrate (Sigma-Aldrich) retrieval buffer (10 mM, pH 6.0). The slides were then cooled to room temperature. Endogenous peroxidase was blocked by incubating the slides in a solution of 1% (*v*/*v*) hydrogen peroxide (30%, Merck) in methanol. The sections were blocked in 3% (*v*/*v*) normal goat serum (Sigma-Aldrich) in 0.1% (*v*/*v*) Tween-20/PBS for 1 h at room temperature in a humidified chamber. Avidin/biotin blocking was performed using a kit (Abcam, Cambridge, UK) as per manufacturer’s instructions. The primary antibody was prepared in blocking solution (3% (*v*/*v*) normal goat serum in 0.1% (*v*/*v*) Tween-20/PBS), added on to the sections, and incubated for 16 h at 4 °C in a humidified chamber. Slides were then washed gently in 0.1% (*v*/*v*) Tween-20/PBS for 3 min three times. The secondary antibody was added onto the sections and incubated for 1 h at room temperature in a humidified chamber. During this incubation period, the avidin-biotin-complex (ABC) solution (Vector Laboratories, Newark, CA, USA) was prepared as per manufacturer’s instructions and incubated for 30 min at room temperature. The slides were washed twice in 0.1% (*v*/*v*) Tween-20/PBS followed by twice in PBS for 3 min per wash. The ABC solution was added onto the sections and incubated for 1 h at room temperature in a humidified chamber. The slides were washed for 3 min, twice in 0.1% (*v*/*v*) Tween-20/PBS followed by twice in PBS for 3 min. The 3,3′-diaminobenzidine (DAB, Vector Laboratories) substrate solution was prepared as per manufacturer’s instructions. The DAB solution was added onto the sections and developed by observing under a light microscope until a prominent brown staining appeared and slides placed in distilled water to prevent further signal development. A counterstain was performed as follows: distilled water, 1 min; dipped in haematoxylin (Amber Scientific); distilled water, 1 min; Scotts tap water substitute (Amber Scientific), 2 min; distilled water, 1 min; 100% ethanol, 30 s; 100% ethanol, 1 min; 100% ethanol, 2 min; and histolene, 4 min (three times). Cover slips were then applied. Antibody combinations used in IHC are outlined in Table 2.

### 2.12. Pathological Grading of MMTV-PyMT and MMTV-PyMTxHPSE^−/−^ Mammary Tumour Development

Mammary glands stained with H&E were first analysed to determine the stage of mammary tumour development. Based on the appearance of the ductal structures, an initial pathological grading of normal, hyperplasia, DCIS, or invasive carcinoma was assigned to each mammary gland. In order to further validate the grading obtained by H&E staining, an anti-Ki67 staining was carried out to distinguish grades of hyperplasia, DCIS, and invasive carcinoma from normal mammary ducts. Anti-SMMHC staining was employed to distinguish between DCIS and invasive carcinoma. The total number of mammary glands bearing invasive carcinoma were determined and subjected to Chi-square statistical analysis.

### 2.13. Microvessel Density Quantification following Anti-CD31 IHC

Tissue sections were collected 100–200 μm apart in order to generate a representative histology data stack per tissue sample. Following IHC, photographs were taken under a light microscope. Each image was divided into 10–12 grids, and the visible stained vessels were manually counted. An average vessel count per field was obtained by dividing the total number of visible microvessels counted by the number of fields employed per image.

### 2.14. H-Scoring of Mammary Tumours

H-scoring of anti-MMP-2-stained mouse mammary gland sections bearing DCIS and invasive carcinoma lesions was carried out using the staining intensity guide (Appendix A). A score of 0–3 was assigned to four distinct levels of staining, from no staining to highest intensity. The percentage area of stain for each intensity level within each lesion was then visually determined. The H-score was calculated using the formula [(0 × %area of stain of score 0) + (1 × %area of stain of score 1) + (2 × %area of stain of score 2) + (3 × %area of stain of score 3)]. Statistical analysis was then carried out.

### 2.15. Extraction of Total Lung RNA and Relative Tumour Burden (RTB) Determination by qPCR

Mouse lungs snap-frozen in liquid N_2_ were weighed and homogenised in 1 mL of TRI reagent^®^ (Molecular Research Centre, Inc., Cincinnati, OH, USA) per 100 mg of tissue using a metal bead lysing matrix (MP bio, Santa Ana, CA, USA) and with high-speed disruption using a FAST-PREP-24^TM^ instrument (MP bio). Bromoanisole (Molecular Research Centre, Inc.) was added to the homogenate at a ratio of 1:20, respectively, and shaken vigorously for 15 s. The mixture was centrifuged at 12,000× *g* for 15 min at 4 °C to facilitate phase separation. The upper aqueous phase was extracted and mixed with 1 volume of isopropanol (Sigma-Aldrich), incubated for 10 min at room temperature, mixed vigorously, and centrifuged at 12,000× *g* for 5 min at 4 °C. The supernatant was discarded, and the pellet was washed with 2 volumes of 75% ethanol, mixed vigorously, and centrifuged at 12,000× *g* for 5 min at 4 °C. All ethanol was removed, and the pellet was solubilised in nuclease-free water (New England Biolabs, Ipswich, MA, USA). Prior to cDNA synthesis, the purified RNA was subjected to a DNase I (New England Biolabs) digestion as per manufacturer’s instructions. cDNA was synthesised using the iScript cDNA synthesis kit (Bio-Rad, Hercules, CA, USA), as per manufacturer’s instructions using a thermocycler: 25 °C, 5 min; 42 °C, 30 min; and 85 °C, 5 min. The qPCR master reaction was prepared with a total of 120 ng of DNA and 54.7% (*v*/*v*) Sybr Green (ThermoFisher) with 0.8 μM of each primer (Table 3). This working solution was divided into three to carry out the reaction in triplicate. The qPCR cycle was as follows: one cycle of 95 °C for 10 min, 40 cycles of 95 °C for 30 s, 62 °C for 30 s, 72 °C for 40 s, then 95 °C for 1 min, and a melting curve generation from 55–95 °C with 30 s per 1 °C increment. Relative tumour burden (RTB) in lungs was calculated as previously described using the following formula: RTB = 10,000/2^ΔCq^ where ΔCq = Cq (target gene, PyMT) − Cq (control, 18S) [42].

## 3. Results

### 3.1. Characterisation of HPSE Expression and Activity Status of MMTV-PyMT and MMTV-PyMTxHPSE^−/−^ Mice

The HPSE-null status of MMTV-PyMTxHPSE^−/−^ mice was initially characterised (Figure 1). The lack of HPSE expression and activity in C57Bl/6xHPSE^−/−^ mice which were used in the generation of MMTV-PyMTxHPSE^−/−^ mice (Appendix A) for this study has been previously reported [7]. In order to verify the HPSE expression status of MMTV-PyMT and MMTV-PyMTxHPSE^−/−^ mice, a Con-A sepharose bead pull down assay was performed on whole spleen lysates of female animals. This resulted in a prominent band of approximately 45 kDa upon immunoblotting, which corresponded to the size of purified human HPSE and was present only in the spleen lysates of MMTV-PyMT mice (Figure 1A).

The HPSE enzymatic activity status of the MMTV-PyMTxHPSE^−/−^ mice was then investigated. An enzyme activity assay was performed to determine HPSE activity in whole spleen lysates of female MMTV-PyMT and MMTV-PyMTxHPSE^−/−^ mice [43]. Whole spleen lysate of MMTV-PyMTxHPSE^−/−^ mice exhibited a significantly lower level of HPSE activity compared with that of MMTV-PyMT mice (Figure 1B). In order to determine the expression of HPSE in mammary tumour lesions of MMTV-PyMT and MMTV-PyMTxHPSE^−/−^ mice, an anti-HPSE IHC assay was performed. The fourth inguinal mammary glands were excised from animals euthanised at the ethical cumulative tumour volume end point. Tumour lesions of at least DCIS grade within the mammary glands of MMTV-PyMT mice showed distinct regions of HPSE expression, while those of MMTV-PyMTxHPSE^−/−^ mice showed a lack of such regions (Figure 1C). These results combined confirm the lack of HPSE expression in MMTV-PyMTxHPSE^−/−^ mice.

### 3.2. Evaluation of Spontaneous Mammary Tumour Growth between MMTV-PyMT and MMTV-PyMTxHPSE^−/−^ Mice

Based on current literature, it was hypothesised that female MMTV-PyMTxHPSE^−/−^ mice would exhibit a significantly less-aggressive mammary tumour growth profile in contrast to female MMTV-PyMT mice. Cumulative tumour volumes of female MMTV-PyMT and MMTV-PyMTxHPSE^−/−^ mice were measured from when tumours were first palpable and measurable to when the animals were euthanised at the ethical cumulative tumour volume end point (Figure 2). Mammary tumour growth rates between female MMTV-PyMT and MMTV-PyMTxHPSE^−/−^ mice were thus observed to be comparable (Figure 2A). The time taken to reach the ethical cumulative tumour volume end point was also comparable between female MMTV-PyMT and MMTV-PyMTxHPSE^−/−^ mice (Figure 2B). The tumour latency periods to when palpable and measurable mammary tumours were first detected were also comparable between female MMTV-PyMT and MMTV-PyMTxHPSE^−/−^ mice (Figure 2C). To determine total mammary gland weights at the ethical end point, mammary glands were excised and weighed immediately following euthanisation. No significant difference in gross mammary gland weights was observed between female MMTV-PyMT and MMTV-PyMTxHPSE^−/−^ mice (Figure 2D). To normalise mammary gland weight measurements, female MMTV-PyMT and MMTV-PyMTxHPSE^−/−^ mice were euthanised at 20 weeks of age and total tumour weights were measured. No significant difference was observed in age-matched gross mammary tumour weights (Figure 2E). These results therefore suggest that HPSE has no significant effect on the spontaneous mammary tumour development and growth kinetics in MMTV-PyMT mice.

### 3.3. HPSE Expression over Time in MMTV-PyMT Mammary Glands

Although it is known that HPSE is overexpressed in breast cancer, it is not well understood precisely when during mammary tumour development this change occurs [26]. An attempt was made to determine when significant HPSE expression first became evident during the mammary tumour progression of MMTV-PyMT mice (Figure 3). Initial H&E staining of fourth inguinal mammary glands of MMTV-PyMT mice at 4, 8, 12, 16, and 20 weeks of age confirmed tumour lesions of at least hyperplasia grade. This was followed by an anti-HPSE IHC, which revealed detectable levels of HPSE expression within the lesions occurring as early as 4 weeks of age and remaining consistent over time. No significant overexpression of HPSE associated with the age of the mice was observed. Therefore, these results suggest that in MMTV-PyMT mice, tumour lesion-associated HPSE expression within the mammary glands occurs early in tumour development and remains at a steady level throughout growth.

### 3.4. The Effect of HPSE on Early- and Late-Stage Tumour Angiogenesis in MMTV-PyMT Mice

It is known that HPSE promotes angiogenesis in solid tumours, leading to enhanced tumour growth [22,44,45,46]. To investigate this phenomenon in early- and late-stage mammary tumour development, anti-CD31 IHCs were performed on serial sections of 6-week-old mammary glands and mammary tumours excised at the ethical tumour volume end point, respectively, of MMTV-PyMT and MMTV-PyMTxHPSE^−/−^ animals (Figure 4). Quantitative analysis revealed no significant difference in the number of mammary lesion-associated vessels in mammary glands of 6-week-old MMTV-PyMT and MMTV-PyMTxHPSE^−/−^ animals (Figure 4A,B). However, tumours excised at the ethical tumour volume end point from female MMTV-PyMTxHPSE^−/−^ mice exhibited a significantly reduced level of microvessel density compared with those from MMTV-PyMT mice (Figure 4C,D). These results suggest that HPSE may not play a significant role in angiogenesis during early mammary tumour development in MMTV-PyMT mice but is a critical regulator over time.

### 3.5. The Role of Host HPSE in Influencing Lung Metastasis of MMTV-PyMT Mammary Tumours

Mammary tumours in MMTV-PyMT mice result mainly in lung metastases [47]. Serial lung sections were stained with H&E and visually examined to confirm the presence of metastatic lesions (Figure 5). It was thus revealed that both female MMTV-PyMT and MMTV-PyMTxHPSE^−/−^ mice euthanised at the ethical tumour volume end point did indeed exhibit lung metastases, although only 50% of all mice of either strain presented visible lesions (Figure 5A). A quantitative analysis of the number of visually confirmed metastatic lesions did not reveal a significant difference between female MMTV-PyMT and MMTV-PyMTxHPSE^−/−^ mice (Figure 5B). Further quantitative analysis of the lung metastatic burden between the two strains by calculating lung RTB confirmed the previous findings, showing no significant difference between female MMTV-PyMT and MMTV-PyMTxHPSE^−/−^ mice euthanised at the ethical tumour volume end point (Figure 5C). These results suggest that even though HPSE is known to promote metastasis in numerous cancer settings, HPSE in the MMTV-PyMT mouse model may not play an important role.

### 3.6. Evaluation of the Role of HPSE in the Early Stages of Mammary Tumour Development in the MMTV-PyMT Mouse Model

Although HPSE has been reported to promote the development of breast cancer, the precise role of HPSE in the early stages of mammary tumour development remains poorly defined. Mammary tissue sections from 6-week-old MMTV-PyMT and MMTV-PyMTxHPSE^−/−^ mice were examined to reveal the pathological status of tumour lesions present in each mammary gland (Figure 6). The presence of lesions was first determined through H&E staining, which revealed normal mammary glands or malignant grades of hyperplasia, DCIS, and invasive carcinoma. Further staining with anti-Ki67 and anti-SMMHC antibodies distinguished between hyperplasia, DCIS, and invasive carcinoma stages (Figure 6A) [48]. A graphical representation shows the distribution of incidences of each major stage of breast cancer observed between all mammary glands analysed (Figure 6B). This was further expanded upon to represent the distribution between non-invasive and invasive lesions observed in each of the four mammary glands per animal analysed in this study (Appendix A). No distinct variation in lesion grade between female MMTV-PyMT and MMTV-PyMTxHPSE^−/−^ mice was observed. Finally, a quantitative analysis of the number of mammary glands bearing non-invasive vs. invasive mammary tumour lesions between MMTV-PyMT and MMTV-PyMTxHPSE^−/−^ mice was carried out via a chi-square test, as previously described [49]. This revealed no significant variation in invasive lesion incidence between the two groups (Figure 6C). Together, these data suggest that HPSE does not play a role in promoting the progression of MMTV-PyMT mammary tumours early in their development.

### 3.7. Investigating the Presence of a Compensatory Mechanism of MMP-2 Expression in MMTV-PyMTxHPSE^−/−^ Mouse Mammary Tumour Lesions

Due to the dynamic nature of the ECM, the lack of an ECM-modulatory enzyme such as HPSE is sometimes thought to be compensated for by the upregulation of other ECM remodelling enzymes such as MMPs [50]. However, our previous data suggest otherwise [7]. To determine the status of MMP expression within mammary gland lesions, DCIS/invasive lesions were analysed for MMP-2 expression by IHC (Figure 7A). H-score quantification of MMP-2 expression revealed no significant difference between MMTV-PyMT and MMTV-PyMTxHPSE^−/−^ mice (Figure 7B). This suggests that mammary tumour-associated MMP-2 expression is not influenced by the lack of HPSE in MMTV-PyMTxHPSE^−/−^ mice and a previously proposed compensatory mechanism of upregulated MMP expression may not exist in this model.

## 4. Discussion

HPSE has been associated with breast cancer progression, as supported by extensive in vitro and in vivo data, as well as numerous clinical investigations. A majority of cancer-related deaths are the direct result of metastasis [51]. Breast cancer in particular, proves potentially life-threatening in its metastatic setting in contrast to when localised to its primary site. As a key promoter of tumour growth and metastasis, HPSE has therefore garnered much attention over the past several decades in the context of not only breast cancer but also most other cancer types. Here, we utilised the well-defined MMTV-PyMT mouse model of spontaneous mammary tumour development to define the precise role of HPSE in murine breast cancer progression, which may in turn complement our understanding of the human malignancy.

The study of breast cancer has involved a large variety of mouse models over several decades, whilst acknowledging that the perfect in vivo model does not exist [52,53]. The MMTV-PyMT mouse model is a well-characterised, robust mouse model of spontaneous mammary tumour development and metastasis, which closely resembles the development of human breast cancer [54]. Therefore, it has emerged as an ideal in vivo model to characterise the human malignancy. These are transgenic mice where the polyomavirus middle-tumour antigen (PyMT) expression is driven by the mouse mammary tumour virus (MMTV) promoter’s long terminal repeat (LTR) regulatory sequence [47]. The PyMT is a robust oncogenic antigen with a demonstrated capacity to transform mouse cells and to give rise to cancer [55,56,57,58]. The tumour induction is through the mimicking of an activated growth factor receptor and has been demonstrated to increase cellular responsiveness to growth factors [59,60]. The development of multifocal tumours involves the activation of c-src, PI3-K, and Raf-Mek-ERK signalling pathways, resulting in the malignant transformation of the mammary epithelial cells [61,62,63]. At the time of derivation of these animals, a majority of mice were reported to develop lung metastases [47].

Numerous studies have elucidated the malignant progression in the MMTV-PyMT model. The analysis of gene activation at each distinct stage of tumour development showed a remarkably similar gene expression pattern between both later stages and the beginning of tumour growth, shedding new light on the transcriptional dynamics of this model [64]. Lin and colleagues have described in detail the mammary tumour growth process in MMTV-PyMT mice [54]. Here, it was reported that like humans, mice exhibited morphologically distinct stages of hyperplasia, adenoma, early carcinoma, and late carcinoma. The expression of biomarkers, too, was consistent with human breast cancers resulting in a poor prognosis, with a gradual loss in ER, progesterone receptor, as well as integrin-β1 along with the persistent expression of cyclin-D1 and ErbB2/Neu (equivalent to human HER2). Malignancy was also associated with an increased influx of leukocytes. Myoepithelial cells, critical in limiting metastasis in mammary tumours, as well as proliferation markers were recently utilised in developing a scoring matrix to distinguish normal epithelium, hyperplasia, intraepithelial neoplasia, and invasive carcinoma in MMTV-PyMT mice [48]. This would enable the transition of in vivo findings to a clinical setting.

PyMT-driven mammary carcinogenesis has been shown to upregulate MMP-13 with no promotion of tumour growth [65]. It also promotes the expression of osteopontin, demonstrated to promote metastasis through EMT plasticity regulation [66,67]. However, the influence of PyMT-driven carcinogenesis on HPSE expression has not been described. The MMTV-LTR contains a glucocorticoid hormone response element, resulting in a hormonal-driven regulation of protein expression [68,69,70,71]. These features result in localised, spontaneous tumour growth with associated metastasis. Several mouse strains have been derived based on the MMTV-PyMT model for the study of breast cancer-related processes. These include PI3K signalling, CSF-1 driven tumorigenesis, the regulation of pulmonary metastasis of mammary carcinoma by CD4^+^ T-cells, the role of neutrophils in supporting lung metastasis in breast cancer, the CD44-mediated metastatic invasion during breast cancer, VEGF-mediated mammary tumour growth, TGF-β signalling, urokinase-mediated breast cancer metastasis, and the role of mitogen-activated protein kinase kinase kinase-1 in breast cancer metastasis [72,73,74,75,76,77,78,79,80]. Thus, the MMTV-PyMT model has established itself as an ideal tool in the study of breast cancer.

Most studies on the role of HPSE in breast cancer have been conducted using in vivo models incorporating HPSE overexpression, the use of isolated breast cancer cells in vitro, or the use of human clinical samples. No studies thus far have reported the use of an in vivo model of spontaneous mammary tumour development incorporating the genetic ablation of HPSE expression. Thus, we have aimed to address the current gap in knowledge with the use of MMTV-PyMTxHPSE^−/−^ mice.

Previously, we had established a C57Bl/6xHPSE^−/−^ mouse strain and demonstrated the lack of HPSE expression in these animals both at a transcription level and a protein expression level [7]. Following the generation of the MMTV-PyMTxHPSE^−/−^ mice described herein, it was necessary to also demonstrate the lack of HPSE expression localised specifically to regions of tumour lesions in the mammary glands of female MMTV-PyMTxHPSE^−/−^ mice. As expected, no HPSE expression was observed within DCIS/invasive mammary tumour lesions of female MMTV-PyMTxHPSE^−/−^ mice. Based on published literature, female MMTV-PyMTxHPSE^−/−^ mice were hypothesised to develop less aggressive mammary tumours in contrast to MMTV-PyMT mice. However, our results suggest otherwise. Mammary tumours in both female MMTV-PyMT and MMTV-PyMTxHPSE^−/−^ mice progressed in a similar manner with similar tumour latency. No single parameter used to evaluate mammary tumour growth suggested a disadvantage of lacking HPSE expression in these animals. These data therefore suggest an HPSE-independent mode of mammary tumour development in the MMTV-PyMT mouse model.

Recently, the MMTV-directed overexpression of HPSE in the mammary glands of mice demonstrated an HPSE-mediated promotion of mammary gland development [27]. In contrast, our present study suggested no distinct role for HPSE in mammary gland branching morphogenesis of mammary glands examined at the ethical tumour volume end point [27,50]. It should be noted, however, that the studies by Boyango et al. stimulated the overexpression of HPSE specifically in the mouse mammary glands and compared HPSE-transgenic mice to their wild-type counterparts. An HPSE-deficient control was not employed in arriving at this conclusion. It should further be noted that mouse mammary glands undergo ovarian cycle-dependent morphological changes, and its development is strongly hormonal-driven [81,82,83]. Thus, it may be crucial to consider synchronising the oestrous cycle status of female mice used in similar studies in order to better define the role of HPSE independently of the effects of varying hormone levels. Estrogen is a key driver of mammary tumorigenesis [84]. Previous studies have indicated ER-driven HPSE expression in breast cancer [24]. As ER is gradually lost in the MMTV-PyMT mouse model over time, this may impact overall HPSE activity in the mammary TME [54]. This relationship, although beyond the scope of our study, warrants further investigation.

Angiogenesis is a key hallmark of tumour progression, and HPSE has been well characterised as a promoter of neovascularisation [22]. Consistent with this notion, mammary tumours excised from female MMTV-PyMTxHPSE^−/−^ mice at the cumulative ethical tumour volume end point were shown to possess significantly reduced vasculature in contrast to those from MMTV-PyMT mice. This observation aligns with those of previous studies conducted using both in vivo and human clinical samples [22,85]. However, no significant variation in mammary tumour lesion-associated microvessels was observed in the early stages of MMTV-PyMT mammary tumour growth. These observations suggest that HPSE is a promoter of angiogenesis in MMTV-PyMT mice, with effects most evident in developed tumours at the ethical tumour volume end point. However, it is interesting to note that reduced angiogenesis in female MMTV-PyMTxHPSE^−/−^ mice did not result in impaired mammary tumour growth. The current literature reports an increase in HPSE expression leading to increased angiogenesis that in turn correlates with an increased tumour size and growth rate in numerous cancer settings as discussed earlier. Based on the observations reported herein, it can be suggested that in the MMTV-PyMT model, a reduction in mammary tumour angiogenesis as a result of the lack of HPSE expression does not impact the overall tumour growth capacity. A similar relationship in human malignancies is yet to be reported. Indeed, these data suggest that in certain clinical settings, the use of HPSE inhibitors aimed at inhibiting tumour growth through impaired angiogenesis may not prove to be effective. This concept requires further investigation.

It should also be noted that angiogenesis is not dependent on HPSE but rather enhanced through its enzymatic activity. Therefore, it is likely that the inherent capacity of a mammary tumour to generate its vasculature independently of HPSE activity is sufficient to maintain tumour growth. This likely occurs through the activity of enzymes such as MMPs, with ECM-modulating and angiogenesis-inducing capabilities [86]. It also remains to be seen if the reduction in angiogenesis in HPSE-devoid mammary tumours leads to increased hypoxia and in turn, leads to downstream HIF-mediated survival pathways, resulting in the maintenance of tumour progression [87,88].

These data gathered from investigating primary tumour angiogenesis in MMTV-PyMT mice further suggest that a distinct overexpression of HPSE may likely be observed at a specific stage of MMTV-PyMT mammary tumour development. However, our efforts to define the kinetics of mammary tumour lesion-associated HPSE expression suggest that HPSE expression remains at a consistent level throughout mammary tumour development in MMTV-PyMT mice. It should be noted that the high lipid content of the mammary glands prevented the conduct of enzymatic activity assays to demonstrate the HPSE activity within these glands over time, and therefore, an anti-HPSE IHC was employed. Combined with the previous data on HPSE-driven tumour angiogenesis, these observations suggest that a distinct upregulation of HPSE expression may not occur in MMTV-PyMT mice during mammary tumour development, but a consistent level of HPSE expression is sufficient to activate the angiogenic switch.

Spontaneous metastasis is a key feature of the MMTV-PyMT mouse model, which further warrants its use in the understanding of breast cancer progression. The use of MMTV-PyMTxHPSE^−/−^ animals therefore enabled the investigation of the influence of HPSE expression on breast cancer metastasis. As discussed previously, tumour growth and metastasis are significantly affected by the immune system. Key components of the immune system such as NK cells have been demonstrated to be pivotal in anti-tumour immunity and additionally, to be dependent on intracellular HPSE activity in order to exert cytotoxic effects [6,12]. Based on this and other observations previously elaborated upon, a significantly higher lung tumour burden was expected in mammary tumour-bearing female MMTV-PyMTxHPSE^−/−^ mice at the ethical tumour volume end point, in contrast to MMTV-PyMT mice. However, our findings indicate that lung metastasis in tumour-bearing female MMTV-PyMT mice remained unaffected by the lack of HPSE expression in the host tissue. It is possible that a difference in lung metastatic burden between mammary tumour-bearing MMTV-PyMT and MMTV-PyMTxHPSE^−/−^ mice exists in the early stages of tumour growth that is not evident at the ethical tumour volume end point. However, it should be noted that due to the extremely low level of metastatic dissemination in the animals used, analysis by qPCR to determine a difference in lung RTB at an early stage prior to the ethical tumour volume end point may prove unfeasible.

It should also be noted that MMTV-PyMT transgenic mice have been generated on a variety of genetic backgrounds, which in turn has a significant impact on mammary tumour metastasis. It has been demonstrated that MMTV-PyMT mice on an FVB background are significantly more susceptible to metastatic disease compared with those on a C57Bl/6 background [89]. Indeed, the initial reporting of the MMTV-PyMT strain by Guy et al. employed FVB animals, where nearly all tumour-bearing MMTV/middle T transgenic animals developed metastases [47]. However, our studies employed transgenic mice on a C57Bl/6 background. Therefore, it should be noted that only 50% of animals employed in the studies reported herein exhibited histologically detectable lung metastases. A follow-up analysis of PyMT expression using qPCR further confirmed this observation. Future studies focusing on the role of HPSE in mediating mammary tumour metastasis in MMTV-PyMT transgenic mice may benefit from using animals generated on an FVB background. Attempts were made to investigate the role of HPSE in the early stages of mammary tumour development, as the current literature is primarily focused on disease in its later stages. The actions of ECM-modulating enzymes such as HPSE may have a profound impact especially in the early stages of tumour development, translating to clinically obvious effects in advanced stages of cancer. Despite this potential, no study has thus far reported as such. The use of the spontaneous mammary tumour-developing MMTV-PyMT mice provided an excellent model to understand the role of HPSE in promoting early mammary tumour establishment. A visual examination was first carried out on serial sections of mammary glands. It was important to isolate mammary glands at an age when distinct stages of tumour development would be present, without an overwhelming presence of invasive carcinoma lesions. Guidelines were adapted from studies previously conducted by Duivenvoorden et al. with 6 weeks of age determined as the ideal stage to observe variations in early mammary tumour establishment between female MMTV-PyMT and MMTV-PyMTxHPSE^−/−^ mice [48,49]. Although initial H&E staining of the mammary gland sections confirmed the pathology of carcinoma or the absence of lesions to a certain degree, it was necessary to follow up with IHC confirmation. This was due to subtle variations between tumour grades not being visible upon basic histological examination. To distinguish normal mammary ducts from those undergoing rapid epithelial proliferation and progressed to a stage of hyperplasia, an anti-Ki67 IHC was employed [48]. This identified epithelial cells undergoing rapid division, indicating early stages of carcinoma development. To distinguish between DCIS and invasive carcinoma, an anti-SMMHC IHC was employed [48]. This in turn enabled the identification of intact ductal walls of DCIS lesions from invasive carcinoma, which showed a severe disruption or a complete lack of ductal walls. However, statistical analysis of the number of mammary glands exhibiting invasive carcinoma versus those that did not suggest no significant difference in the promotion of mammary tumour invasion between female MMTV-PyMT and MMTV-PyMTxHPSE^−/−^ mice at 6 weeks of age. No distinct pattern in the incidence of invasive carcinoma between the different mammary glands studied was observed, neither was a significant variation between each of the four major stages of mammary tumour development observed between female MMTV-PyMT and MMTV-PyMTxHPSE^−/−^ mice. These results in MMTV-PyMT mice collectively suggest that HPSE imparts no effects in the early stages of mammary tumour development but may play a role in later stages. However, it remains to be determined if these findings are relevant in the context of human breast cancer. Further IHC studies to investigate a relationship between HPSE expression and epithelial proliferation as indicated via Ki67 expression may shed light on any HPSE-driven proliferative mechanisms during early mammary tumour development. Furthermore, this study could be complemented with an analysis of CD31 expression to explore the influence of HPSE expression on cellular proliferation and associated angiogenesis.

The ECM is a complex, dynamic environment [90,91,92]. Indeed, the multitude of ECM components with sometimes overlapping functions suggests the likelihood that the lack or inhibition of one component may be compensated for by another with a similar role within the ECM. In certain other cases, the targeted inhibition of an ECM component with the aim of curbing the severity of disease may lead to unintended side effects, as the same component could be vital in maintaining tissue homeostasis. Previously, an upregulation of MMPs was described in HPSE-deficient mice in response to the lack of HPSE expression [50]. This indicated a co-regulatory mechanism between MMPs and HPSE in the ECM and indicated an upregulation of MMP-2 expression in the mammary glands. These observations could lead to the assumption that an upregulation of MMP expression in female MMTV-PyMTxHPSE^−/−^ mice may explain the results of the studies reported herein, which suggested an HPSE-independent mode of mammary carcinoma progression in the MMTV-PyMT mouse model. However, upon the generation of the C57Bl/6xHPSE^−/−^ animals that formed the founding members of the MMTV-PyMTxHPSE^−/−^ mice reportedherein, the expression levels of a range of MMPs (MMP-2, -9, -14, and -25) in a variety of tissues were analysed [7]. Here, it was reported that MMP expression levels in HPSE-deficient mice remained unchanged. In order to further elaborate on the observations by Poon et al. and to investigate if indeed an MMP-2 overexpression was present in the mammary glands of female MMTV-PyMTxHPSE^−/−^ mice as described by Zcharia et al., an anti-MMP-2 IHC was undertaken. Mammary tumour lesions of DCIS/invasive grade were chosen for this purpose as the previous studies employed qPCR, which had no means of pinpointing if MMP upregulation occurred within the lesions itself. Aligned with our previous findings, no significant difference in MMP-2 expression was observed in MMTV-PyMTxHPSE^−/−^ mice. Further studies may be undertaken to complement these histological findings via qPCR by investigating the expression of MMP-2 and other relevant MMPs such as MMP-14 in whole mammary tumours [93,94].

## 5. Conclusions

This study explored the effects of HPSE on the spontaneous tumour growth of MMTV-PyMT mice and identified that the mammary tumour progression in these animals occurred in an HPSE-independent manner. Although primary tumour angiogenesis was affected by the lack of HPSE, the overall tumour burden, tumour growth rate, and metastasis in mammary tumour-bearing MMTV-PyMT mice remained unperturbed. HPSE was also suggested to play no significant role during the early stages of mammary tumour development, which had remained largely undefined. Furthermore, no compensatory mechanism of MMP-2 overexpression to counter the lack of HPSE expression was observed.

Collectively, our data suggest that in some human breast cancer settings and possibly in other cancer settings, HPSE may not always play as significant a role as predicted based on published literature. This could have significant implications in the current development and validation of HPSE inhibitors for use in clinical settings. Significantly, the newly uncovered ‘double-edged’ nature of HPSE within the TME must be carefully considered when designing HPSE-targeted therapeutics. Recent studies have indicated that HPSE is critical in maintaining the function of the components of the immune system [6,7,12,13,14]. With regard to the balance between the pro-tumorigenic hallmarks of cancer and the anti-tumour components of the immune system, the indiscriminate targeting of HPSE within the TME may tip the scales in favour of promoting the hallmarks of cancer [18].

It is therefore vital that future studies dissect and define the precise role of HPSE within the TME in each tumour setting, as individual cancer types and tumour settings are vastly different from each other. The dual role of HPSE and indeed its potential redundancy in some tumour settings add a significant layer of complexity to this task, but in order to minimise or prevent undesirable off-target effects seen in the past through the indiscriminatory targeting of MMPs, such detailed studies are necessary [95,96]. These considerations will pave the way for important future research to better define the complicated nature of HPSE and its role in tumour progression.

## Figures and Tables

**Figure 1 cancers-15-03062-f001:**
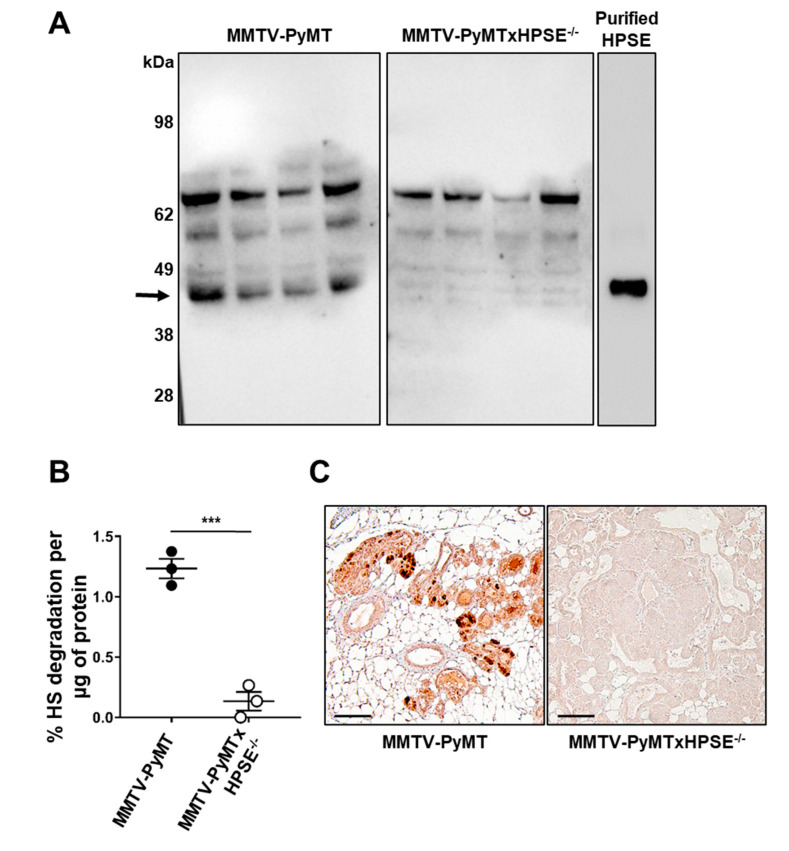
**Confirmation of HPSE^−/−^ status in MMTV-PyMT^−/−^ mice.** (**A**) Con-A-Sepharose bead pull down assay of whole spleen lysates of MMTV-PyMT and MMTV-PyMTxHPSE^−/−^ mice followed by a Western blot assay indicated a prominent band of approximately 45 kDa (arrow) corresponding to enzymatically active HPSE seen only in MMTV-PyMT animals *(n* = 4). (**B**) HS degradation assay with whole spleen lysates of MMTV-PyMT and MMTV-PyMTxHPSE^−/−^ mice showed a significant reduction in HPSE enzymatic activity in MMTV-PyMTxHPSE^−/−^ mice (*n* = 3). (**C**) A representative image of an anti-HPSE IHC of mammary gland sections isolated from MMTV-PyMT and MMTV-PyMTxHPSE^−/−^ mice shows a lack of HPSE expression within DCIS/invasive lesions of MMTV-PyMTxHPSE^−/−^ mice. Scale bar = 100 μm; ***, *p* < 0.001, unpaired *t*-test. The uncropped blots are shown in Appendix A.

**Figure 2 cancers-15-03062-f002:**
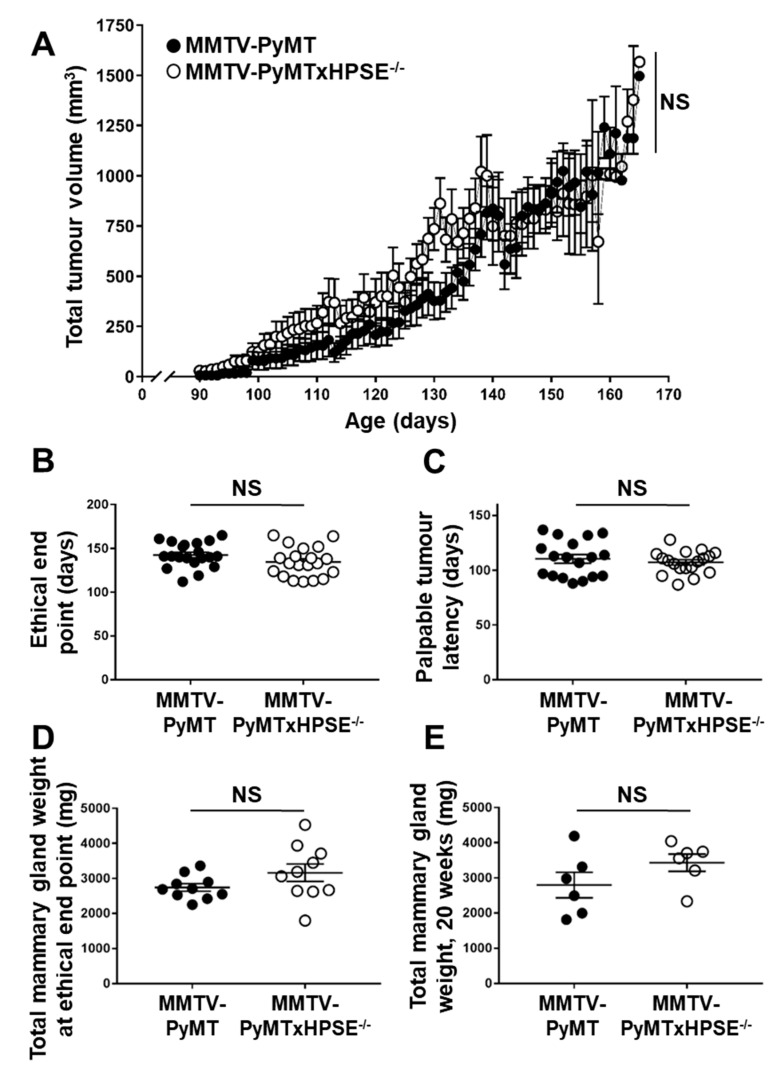
**Evaluation of spontaneous mammary tumour growth between MMTV-PyMT and MMTV-PyMTxHPSE^−/−^ mice.** (**A**) Total mammary tumour growth rates between MMTV-PyMT and MMTV-PyMTxHPSE^−/−^ mice were comparable; *n* = 20 per group. (**B**,**C**) Time to reach the ethical tumour volume end point in MMTV-PyMT and MMTV-PyMTxHPSE^−/−^ was comparable (*n* = 20), as well as the time taken to develop palpable tumours (*n* = 18–20). (**D**,**E**) Total mammary gland weights of MMTV-PyMT and MMTV-PyMTxHPSE^−/−^ at the ethical tumour volume end point (*n* = 10), as well as at 20 weeks of age (*n* = 6) were comparable. Error bars = SEM; NS, not significant; unpaired *t*-test.

**Figure 3 cancers-15-03062-f003:**
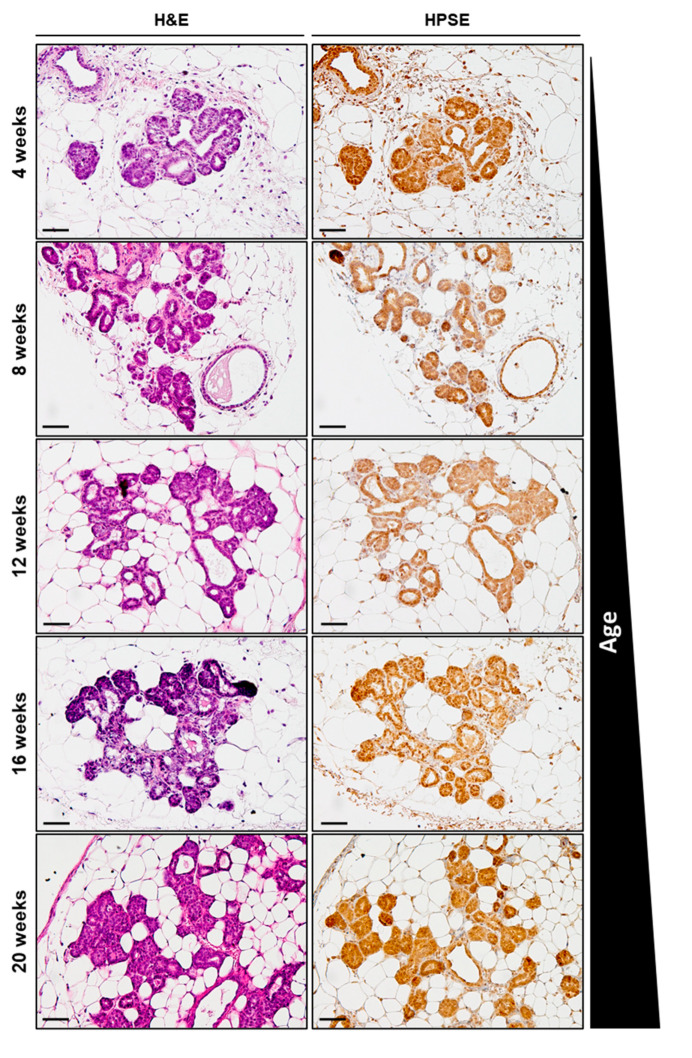
**HPSE expression over time in mammary glands of MMTV-PyMT mice.** Evaluation of HPSE expression by IHC in the 4th inguinal mammary glands of 4-, 8-, 12-, 16-, and 20-week-old MMTV-PyMT mice bearing hyperplastic lesions revealed a consistent level of HPSE expression over time. Representative images of *n* = 3 per age group. Scale bar = 50 μm.

**Figure 4 cancers-15-03062-f004:**
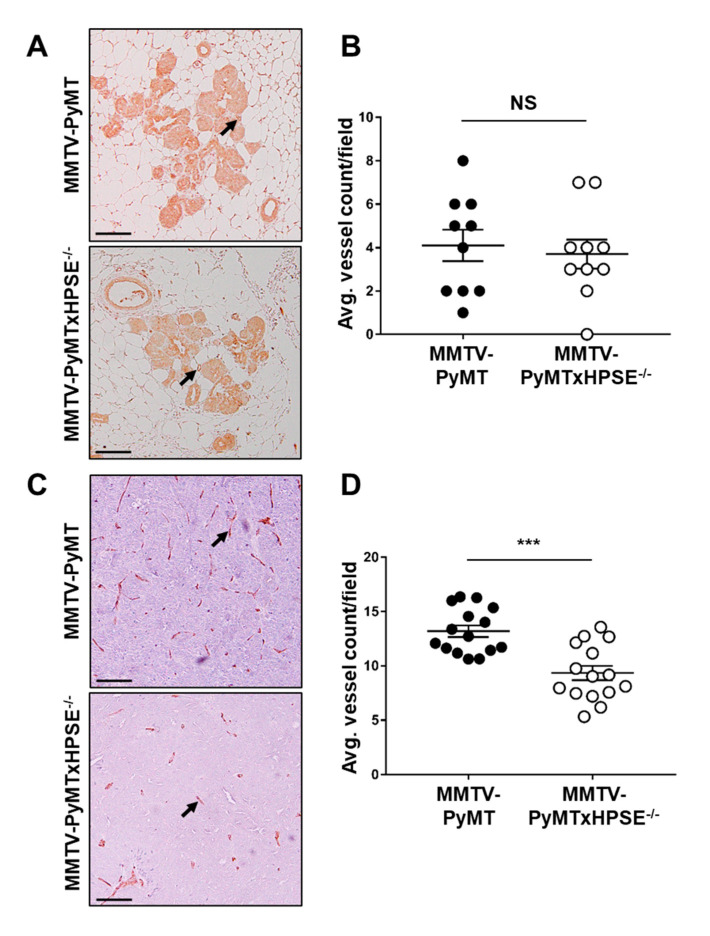
**The role of HPSE in promoting early- and late-stage mammary tumour angiogenesis in MMTV-PyMT mice.** (**A**) Representative anti-CD31 IHC images of DCIS/invasive lesion-bearing mammary glands of 6-week-old MMTV-PyMT and MMTV-PyMTxHPSE^−/−^ mice. Arrows indicate CD31-positive microvessels. (**B**) Visual quantification of blood vessels associated with regions of DCIS and invasive lesions revealed no significant difference in angiogenesis during early tumour development. Pooled data, *n* = 5 per group; 2 sections per mammary gland. (**C**) Representative anti-CD31 IHC images of serial sections of primary mammary tumours excised from MMTV-PyMT and MMTV-PyMTxHPSE^−/−^ mice at the ethical tumour volume end point. Arrows indicate CD31-positive microvessels. (**D**) Quantification of microvessels revealed significantly reduced angiogenesis in late-stage tumours excised from MMTV-PyMTxHPSE^−/−^ mice. Pooled data, *n* = 3 per group; 5 serial sections per tumour. Scale bars = 100 μm. Error bars = SEM; NS, not significant; ***, *p* < 0.001; unpaired *t*-test.

**Figure 5 cancers-15-03062-f005:**
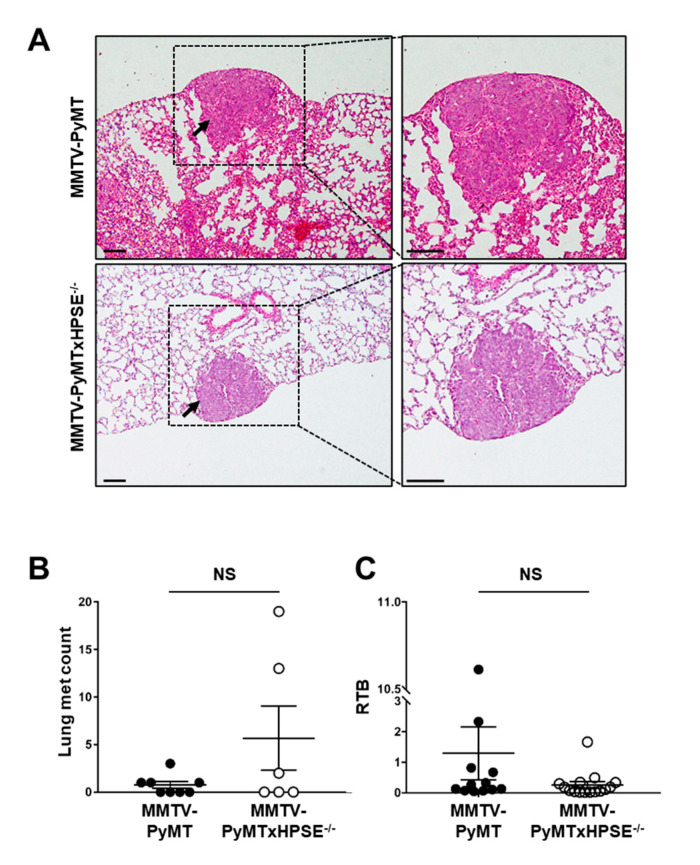
**Lung metastatic burden in MMTV-PyMT and MMTV-PyMTxHPSE^−/−^ mice.** (**A**) Representative images of metastatic lesions (indicated by arrows) in MMTV-PyMT and MMTV-PyMTxHPSE^−/−^ female mice lungs excised at the ethical tumour volume end point; H&E stained. (**B**) Visual quantification of individual metastatic lesions in serial lung sections revealed no significant difference between MMTV-PyMT and MMTV-PyMTxHPSE^−/−^ mice (*n* = 6–8). (**C**) qPCR of MMTV-PyMT and MMTV-PyMTxHPSE^−/−^ mouse lungs excised at the ethical tumour volume end point revealed no significant difference in RTB (*n* = 12–15). Scale bars = 100 μm. Error bars = SEM; NS, not significant; unpaired *t*-test.

**Figure 6 cancers-15-03062-f006:**
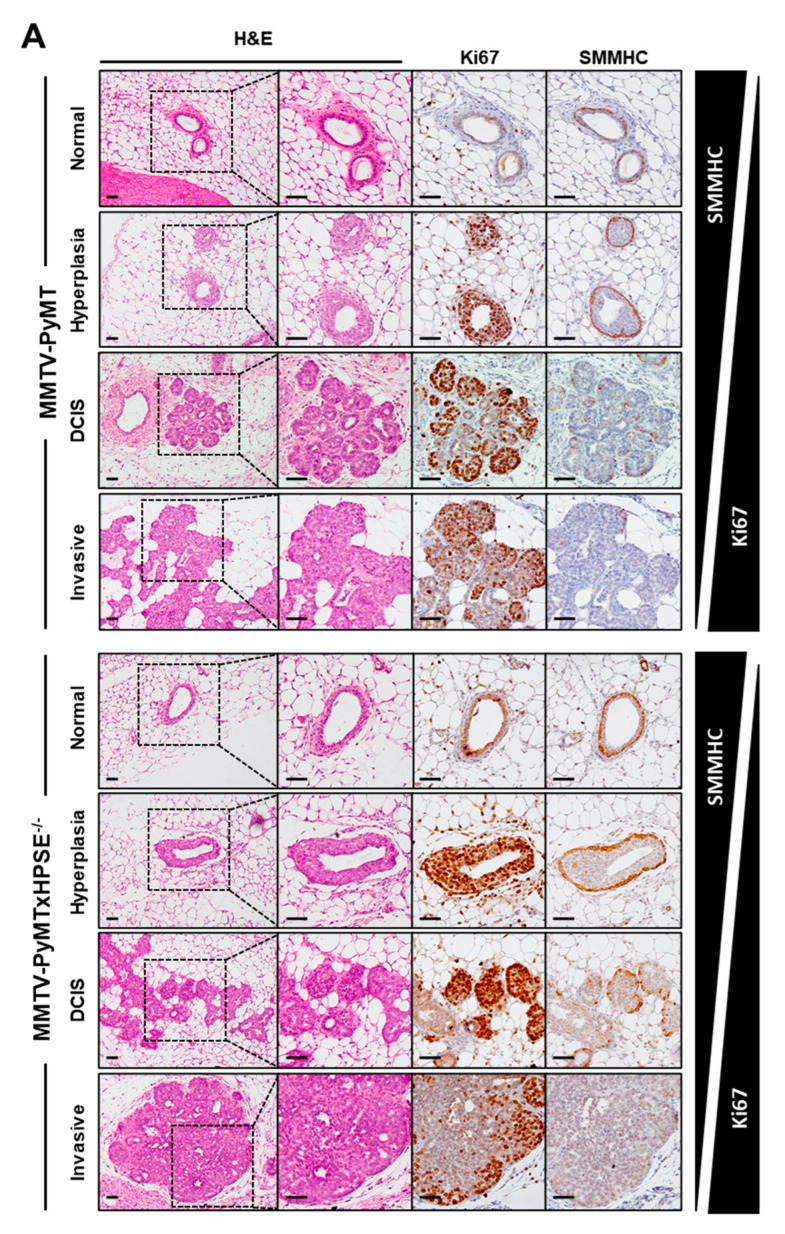
**The role of HPSE in early mammary tumour development in MMTV-PyMT and MMTV-PyMTxHPSE^−/−^ mice.** (**A**) Representative images of normal, hyperplastic, DCIS, and invasive carcinoma regions of MMTV-PyMT and MMTV-PyMTxHPSE^−/−^ mammary glands. H&E staining along with IHC for Ki67 and SMMHC expression confirmed each stage of mammary tumour development. The expression of Ki-67 increases with the progression of carcinoma while that of SMMHC decreases. (**B**) A pie chart representation of the number of mammary glands with each distinct stage of tumour development (represented as parts of 32; *n* = 8 per group, 4 mammary glands per mouse). (**C**) Chi-square analysis of invasive vs. non-invasive phenotypes between mammary glands of MMTV-PyMT and MMTV-PyMTxHPSE^−/−^ mice showed no significant difference (*n* = 8). Scale bars = 50 μm; NS, not significant.

**Figure 7 cancers-15-03062-f007:**
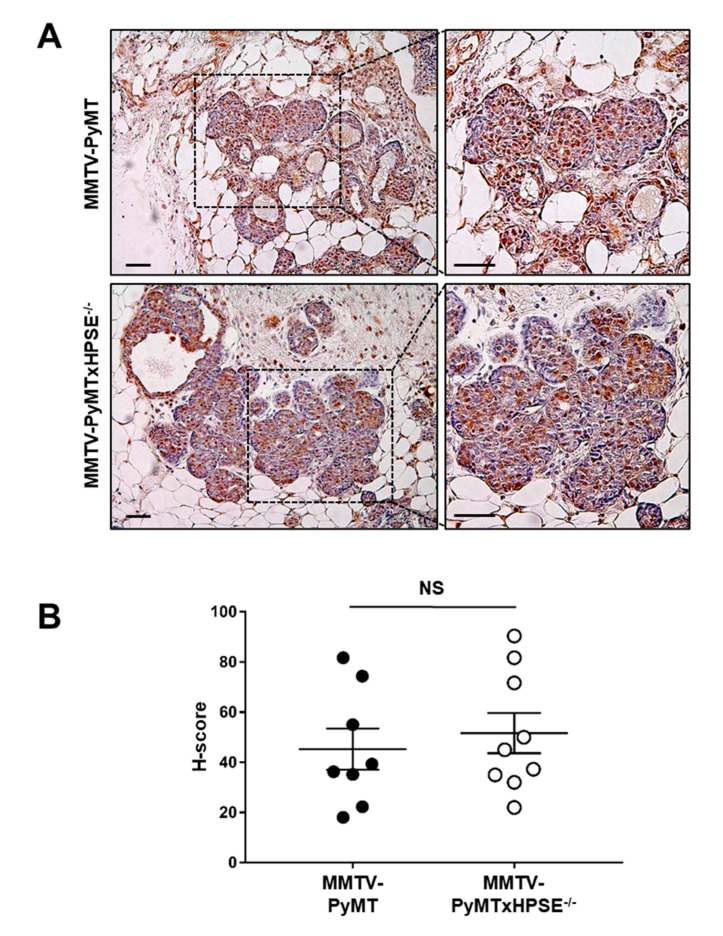
**H-score quantification of MMP-2 expression in DCIS/invasive lesions of MMTV-PyMT and MMTV-PyMTxHPSE^−/−^ mouse mammary glands.** (**A**) Representative MMP-2 IHC images of serial sections of DCIS/invasive lesion-bearing mammary glands excised from 20-week-old MMTV-PyMT and MMTV-PyMTxHPSE^−/−^ mice. (**B**) H-score analysis revealed no significant difference in staining intensity and, therefore, MMP-2 expression levels in DCIS/invasive lesions of MMTV-PyMT and MMTV-PyMTxHPSE^−/−^ mice (*n* = 3). Scale bar = 200 μm; Error bars = SEM; NS, not significant; unpaired *t*-test.

**Table 1 cancers-15-03062-t001:** Genotyping oligonucleotide primers.

Primer	Sequence (5′ to 3′)	Size (bp)
MMTV-PyMT (F)	AGG AAC CGG CTT CCA GGT AAG A	
MMTV-PyMT (R)	TTG GTG TTC CAA ACC ATT GCA T	260
HPSE^+/+^ (F)	GAA GAA CCA TTA TTC ATC TTG CT	
HPSE^+/+^ (R)	CCA AGT GCC AGT CTG CAA GT	143
HPSE^−/−^ (F)	GGG ATG GAT GCA GGT CTT C	
HPSE^−/−^ (R)	CAG ATG GGT GCA GAT TAG ATA T	300
Fabpi (F)	TGG ACA GGA CTG GAC CTC TGC TTT CCT AGA	
Fabpi (R)	TAG AGC TTT CGG ACA TCA CAG GTC ATT CAG	200

**Table 2 cancers-15-03062-t002:** Antibodies used in IHC.

Target	Primary Antibody	Secondary Antibody
HPSE	Rabbit polyclonal anti-HPSE (10 μg/mL, B85543, Abcam)	Biotinylated goat anti-rabbit IgG (H+L) (6 μg/mL, BA-1000, Vector Laboratories)
CD31	Rabbit polyclonal anti-CD31 (16 μg/mL, AB28364, Abcam)	Biotinylated goat anti-rabbit IgG (H+L) (6 μg/mL, BA-1000, Vector Laboratories)
Ki67	Rabbit polyclonal anti-Ki67 (1 μg/mL, AB15580, Abcam)	Biotinylated goat anti-rabbit IgG (H+L) (6 μg/mL, BA-1000, Vector Laboratories)
Smooth muscle myosin heavy chain (SMMHC)	Rabbit monoclonal anti-SMMHC (clone EPR5335, 1.2 μg/mL, AB124679, Abcam)	Biotinylated goat anti-rabbit IgG (H+L) (6 μg/mL, BA-1000, Vector Laboratories)
MMP-2	Rabbit polyclonal anti-MMP-2 (2 μg/mL, AB37150, Abcam)	Biotinylated goat anti-rabbit IgG (H+L) (6 μg/mL, BA-1000, Vector Laboratories)
Normal rabbit IgG (isotype control)	IgG from rabbit serum (various working concentrations, 18140, Sigma-Aldrich)	Biotinylated goat anti-rabbit IgG (H+L) (6 μg/mL, BA-1000, Vector Laboratories)

**Table 3 cancers-15-03062-t003:** Oligonucleotide primers used in determining RTB.

Primer	Sequence 5′ to 3′
PyMT cDNA (F)	CCA ACA GAT ACA CCC GCA CAT
PyMT cDNA (R)	GGT CTT GGT CGC TTT CTG GAT A
Control 18S cDNA (F)	GTA ACC CGT TGA ACC CCA TT
Control 18S cDNA (R)	CCA TCC AAT CGG TAG TAG CG

## Data Availability

All data has been presented in this study.

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
