# Peer review of "Investigating the Role of Heparanase in Breast Cancer Development Utilising the MMTV-PyMT Murine Model of Mammary Carcinoma"

_cancers, 2023, doi:10.3390/cancers15113062_

Round 1

Reviewer 1 Report

The current manuscript entitled “Investigating the role of heparanase in breast cancer development utilising the MMTV-PyMT murine model of mammary carcinoma” (Jayatilleke et al) investigated roles of heparanase in carcinogenesis, growth, and invasion of spontaneously generated tumor in MMTV-PyMT model.  Although the results unexpectedly suggested dispensability of heparanase in the current MMTV-PyMT system, except for angiogenesis (significantly larger number of CD31+ vessels) at the end point. Overall, the current research has been carefully designed and the experiments have conducted with proper quality.   Comments   1) The tumor at the end point from MMTV-PyMT HPSE-/- mice exhibited a significantly reduced level of microvessel density (Figure 4). The authors discussed that this type of angiogenesis is dispensable for the proliferation of cancer in the mammary gland as shown in Figure 3D. The reviewer just wonders this point can be further discussed with the data of Ki67 staining shown in Figure 6A, in which peripheral area of the tumor are restrictedly positive for Ki67 staining in MMTV-PyMT HPSE-/-, whereas whole tumor area are positive in MMTV-PyMT (WT) mice. It looks like that the way of proliferation was somewhat different but Figure 6A lacks CD31+ staining pattern. It is not clear whether the images of  “invasive” cells in Figure 6A were at the end point.   2) The authors examined possible compensatory expression of MMPs, especially MMP2 in the histological analysis (Figure 7). Histological evaluation has difficulty to quantitatively determine the expression level. Also, it is not clear whether what kind of MMP2 are targeted in the experiment, whether intracellular or released, latent form or active form, etc. As stated in the discussion, compensatory expression of other MMPs, especially MT1-MMP (J Cell Biochem 116(8):1668-79 (2015), Dev Cell 47:145-160 (2018)), should be tested. It is likely that the authors prepared total RNA from lung. Similar qPCR studies using RNA from mammary glands would be much easier to check the expression.

Author Response

We thank the reviewer for their positive comments and helpful suggestions. We have addressed the key points raised with the addition of new text to the Discussion section as outlined below. We believe the modifications have improved the manuscript and again thank the reviewer for their help.

Response to comments:

1. The reviewer highlights that the tumors at the end point from MMTV-PyMT HPSE-/- mice exhibit a significantly reduced level of microvessel density (Fig 4) and asks whether this can be further discussed with the data of Ki67 staining shown in Fig 6A. It is difficult to make any definitive conclusion based on these data, however, we have addressed the reviewer's suggestion with the inclusion of new text to the discussion (lines 687-692):

"Further IHC studies to investigate a relationship between HPSE expression and epithelial proliferation as indicated via Ki67 expression may shed light on any HPSE-driven proliferative mechanisms during early mammary tumour development. Furthermore, this study could be complemented with an analysis of CD31 expression to explore the influence of HPSE expression on cellular proliferation and associated angiogenesis."

2. Compensatory expression of MMPs.

The reviewer raises some excellent points on our histological analysis of possible compensatory expression of other MMPs. Regarding the MMP-2 'type' expressed, the IHC approach with the anti-MMP2 antibody employed does not allow discrimination between various expression forms of MMP-2 just a semi-quantitative measure of overall expression levels. We agree with the reviewer that other MMPs may be involved in a compensatory mechanism, such as MT1-MMP, but are not able to conduct such additional experiments for this study as we do not have isolated RNA. However, we have acknowledged this suggestion with the addition of new text and references in the discussion (lines 716-718):

"Further studies may be undertaken to complement these histological findings via qPCR by investigating the expression of MMP-2 and other relevant MMPs such as MMP-14 in whole mammary tumours."

Reviewer 2 Report

Heparanase (HPSE) has been implicated in enhancing the development and progression of solid tumours, including breast cancer. The manuscript entitled “Investigating the role of heparanase in breast cancer development utilising the MMTV-PyMT murine model of mammary carcinoma” by Jayatilleke, K. M. et al. utilise a HPSE-deficient strain of the well-established MMTV-PyMT murine mammary carcinoma model (MMTV-PyMTxHPSE-/- mice) to investigate the role of HPSE in early establishment, progression, and metastasis of mammary tumours. There are several preclinical studies showing that inhibition of HPSE can reduce breast cancer cell proliferation, invasion, and metastasis. On the other hand, there is also evidence suggesting that HPSE may have anti-tumorigenic effects in breast cancer. Therefore, the role of HPSE is still controversial and needs more research to clarify the exactly function of HPSE in the tumor microenvironment context and cancer progression. In this sense the manuscript was appropriate designed and the results was well described, but the authors should include the information of the absence of the estrogen-receptor in this murine model of breast cancer and discuss the association of estrogen-receptor and HPSE on the breast cancer progression. In my opinion the manuscript could be publish in Cancers after minor reviion.

Author Response

We thank the reviewer for their positive comments and the support of our manuscript to be published after minor revision. We agree with the reviewer's suggestion to include 'information of the absence of the estrogen-receptor in this murine model of breast cancer and discuss the association of estrogen-receptor and HPSE on the breast cancer progression". We have added the following new text and references to the discussion lines 577-582 to address this important point:

"Estrogen is a key driver of mammary tumorigenesis [84]. Previous studies have indicated ER-driven HPSE expression in breast cancer [85]. As ER is gradually lost in the MMTV-PyMT mouse model over time, this may impact upon overall HPSE activity in the mammary TME [54]. This relationship, although beyond the scope of our study, warrants further investigation".

Reviewer 3 Report

Authors reported that role of heparanase (HPSE) in MMTV-PyMT murine model to study the breast cancer progression and metastasis. It has been widely studied in various human tissue specific cancers, including breast cancer and inflammation, but limited information with murine models of breast cancer. Upregulation of heparanase associated with enhanced tumor growth, angiogenesis and metastasis in various types of cancer. The results of HPSE in murine model of MMTV-PyMT are contrary to the existing human data. It could be possible to see that sometimes the human and mouse data do not duplicate. Authors clearly demonstrated that deletion of heaparanase in MMTV-PyMT murine model not showed significant effect on early breast cancer progression and metastasis. These findings are important in the clinical setting of HPSE therapy.

Therefore, I recommend this manuscript for publication without any further comments.

Author Response

We thank the reviewer for their positive comments and the support of our manuscript to be published without modification.